# Structural and topological nature of plasticity in sheared granular materials

Yixin Cao[1], Jindong Li[1], Binquan Kou[1], Chengjie Xia[1], Zhifeng Li[1], Rongchang Chen[2], Honglan Xie[2], Tiqiao Xiao[2], Walter Kob [3], Liang Hong[1,4], Jie Zhang[1,4] & Yujie Wang[1,5,6]

Upon mechanical loading, granular materials yield and undergo plastic deformation. The nature of plastic deformation is essential for the development of the macroscopic constitutive models and the understanding of shear band formation. However, we still do not fully understand the microscopic nature of plastic deformation in disordered granular materials. Here we used synchrotron X-ray tomography technique to track the structural evolutions of three-dimensional granular materials under shear. We establish that highly distorted coplanar tetrahedra are the structural defects responsible for microscopic plasticity in disordered granular packings. The elementary plastic events occur through flip events which correspond to a neighbor switching process among these coplanar tetrahedra (or equivalently as the rotation motion of 4-ring disclinations). These events are discrete in space and possess specific orientations with the principal stress direction.

[1] School of Physics and Astronomy, Shanghai Jiao Tong University, 800 Dong Chuan Road, Shanghai 200240, China. [2] Shanghai Institute of Applied Physics, Chinese Academy of Sciences, Shanghai 201800, China. [3] Laboratoire Charles Coulomb, UMR 5521, University of Montpellier and CNRS, 34095 Montpellier, France. [4] Institute of Natural Sciences, Shanghai Jiao Tong University, Shanghai 200240, China. [5] Materials Genome Initiative Center, Shanghai Jiao Tong University, 800 Dong Chuan Road, Shanghai 200240, China. [6] Collaborative Innovation Center of Advanced Microstructures, Nanjing University, Nanjing 210093, China. Correspondence and requests for materials should be addressed to Y.W. (email: yujiewang@sjtu.edu.cn)

Granular solids yield and flow upon applied stress[1,2]. So far, the flow behaviors of granular materials have mainly been treated macroscopically based on empirical constitutive laws[3–5]. More recent approaches treat granular materials within the category of amorphous solids and try to identify the microscopic plastic events to derive the macroscopic mechanical properties[6]. It is generally believed that microscopic plastic events in amorphous solids are induced by certain spatially isolated structural "defects" in the system[7], and the macroscopic yielding, avalanche and shear band formation are induced by their elastic interactions[8]. However, the exact nature of these "defects" remain elusive[9] and it has been investigated based on free volume[10], change of local topology[11], energy landscape[12], shear transformation zones (STZ)[13,14], soft spots as determined by low-energy soft modes[15,16], buckled force chains[17], or defects of an amorphous order[18–21]. Experiments on two-dimensional (2D) soap bubble rafts have identified the elementary plastic event as T1 event which corresponds to two pairs of bubbles switching neighbors with each other[22–24]. Confocal microscopy experiments on three-dimensional (3D) colloidal systems have revealed the elementary plasticity events happening at shear transformation zones with a core radius around three particle diameters[25]. However, the structural basis and topological pathways for these plasticity events have not been investigated in detail. Scattering techniques have also been used to probe local defects in granular systems[26].

In the present study, we carry out quasi-static shear experiments on a 3D disordered granular system, and obtain its structural evolutions by synchrotron X-ray tomography technique (Methods). We find that, similar to T1 event in 2D, the elementary plastic events in 3D are flip events, which consist of two pairs of particles switching neighbors with each other at highly distorted coplanar tetrahedra (structural defects of polytetrahedral order) on Delaunay network. These flip events can equivalently be described as the rotation motions of 4-ring disclination defects in the system and possess specific orientations with the principal stress direction. We therefore establish highly distorted coplanar tetrahedra as dislocation-like structural carriers of plasticity in disordered granular packings, and the flipping processes of them induce plastic deformations.

## Results

**Shear band formation.** Figure 1a is a schematic presentation of the shear cell used in our experiment[27]. Particles inside are monodisperse glass particles (Duke Scientific, $d = 200 \pm 6 \, \mu m$). The samples are prepared with different initial volume fractions $\phi$ and thicknesses $W$ (Supplementary Table 1) and are sheared in the $z$-direction. More details can be found in the Methods section. A tomography scan is carried out after each shear step during which the shear bracket moves up by about $1/3d$. Particle trajectories in the imaging window can be traced during the entire shear process. We define $\Delta r_i (i = x, y, z)$ as the displacement of a particle after a shear step. During the whole shear process, $\Delta r_z$ is the dominant one among all three components. Figure 1b shows $\Delta r_z$ at the beginning of the shear process. The particles with displacement $\Delta r_z > 0.008d$ form a boundary inclined from the vertical direction. This is mainly due to a net positive $\Delta r_x$ component when the system dilates at the beginning of the shear. Upon further shear, when the volume fraction $\phi$ reaches a steady state value, a vertical shear band is formed (Fig. 1c) which can be easily seen either by the spatial distribution of particles with $\Delta r_z > 0.008d$ (Fig. 1c) or the distribution of $\langle \Delta r_z \rangle_x$ along $x$-direction (averaged for all particles located within a $1d$-thickness slice centered at different $x$, Fig. 1d). From Fig. 1d, we estimate the shear band width as $20d$ which is roughly symmetric with respect

to $x = 1d$. The corresponding strain associated with each shear step can therefore be calculated as $\Delta \gamma = \frac{\langle \Delta r_z \rangle_{x=1d}}{10d}$, where $\langle \Delta r_z \rangle_{x=1d}$ is the mean value of $\Delta r_z$ for particles located around $x = 1d$ after each shear step. It turns out $\Delta \gamma = 1.2 \pm 0.2\%$ during the whole-shear process (Supplementary Note 1). The cumulative strain is calculated by $\gamma = \sum \Delta \gamma$ and a total strain $\gamma = 86\%$ is obtained consisting of 71 shear steps for all three samples measured. Despite different initial conditions, each sample reaches steady state forming shear band with similar width and $\phi \sim 0.59$ after a cumulative critical strain $\gamma_c \sim 40\%$ (Fig. 1e). Once steady state is reached, the shear band is in a flow state where the structure continuously relaxes (Supplementary Note 2).

**Non-affine displacement.** Next, we investigated microscopic plastic deformations by the particles' non-affine displacements $\delta r_i (i = x, y, z)$ which we define as $\Delta r_i (i = x, y, z)$ of the particle minus the corresponding average $\langle \Delta r_i \rangle (i = x, y, z)$ of all particles within its radial distance of $2.5d$. It is worth noting that our results are not sensitive to this threshold value or to the particular way non-affine displacement is calculated[14,28] (Supplementary Note 3). Figure 1f, g shows $\delta r_z$ for particles at the beginning of the shear and when the system has reached steady state (only particles having $|\delta r_z| > 0.008d$ are shown), respectively. Contrary to $\Delta r_i$ which is dominated by $\Delta r_z$, $\delta r_i$ have similar magnitudes along all three axes. At the beginning of the shear, there are more particles which have significant non-affine displacements ($|\delta r_z| > 0.008d$) but on average the absolute values are small. In contrast, at the steady state, $|\delta r_z|$ have large absolute values within the shear band (Fig. 1g) and small values outside of it, which is consistent with the general belief that shear band consists of significant plastic activities[1].

**Topology change of local structures.** To understand the structure and topology change upon shear, we partitioned the structural configurations of particles at different shear steps by Delaunay tessellation[20]. We use the parameter $\delta = e_{max} - 1$ to characterize the shape of a tetrahedron, where $e_{max}$, in units of mean particle diameter $d$, is the length of the longest edge of the tetrahedron. A smaller value of $\delta$ suggests that the tetrahedron is closer to a regular one. Upon shear, both structure and topology of the system can vary (Supplementary Note 4). Correspondingly, each tetrahedron can get distorted and eventually destroyed (its vertices no longer belong to the original tetrahedron) which leads to a local topology change. We term a tetrahedron unstable when it is destroyed after a shear step. However, for topological reasons a single tetrahedron cannot get destroyed on its own. In 2D, the topology change follows a specific pathway called T1 event which corresponds to a neighbor switching process[23]. It also corresponds to the destruction of two Delaunay triangles and the subsequent formation of two new ones[23]. In 3D, as shown in Fig. 2a, the topology change happens through pathways called flip events: In a 2–2 flip, two neighboring pairs of coplanar unstable tetrahedra form two new pairs of coplanar tetrahedra by exchanging their vertices, the 2–2 flip is equivalent to its counterpart T1 event in 2D[23]; additionally, a pair of unstable coplanar tetrahedra can also split into three coplanar tetrahedra, or vice versa, which is denoted as 2–3 (or 3–2) flip. The 2–3 (or 3–2) flip, despite its topological significance, could be considered to be only an intermediate step of 2–2 flip in our system, since a consecutive 2–3 and 3–2 flip will yield a 2–2 flip (Fig. 2a) and in reality they almost always happen successively. This is due to the fact that the transient structure is mechanically very unstable. In Fig. 2b, we also analyze the spatial distribution of flip events by showing the cluster size distribution of unstable tetrahedra in space (unstable tetrahedra which are face-adjacent to each other

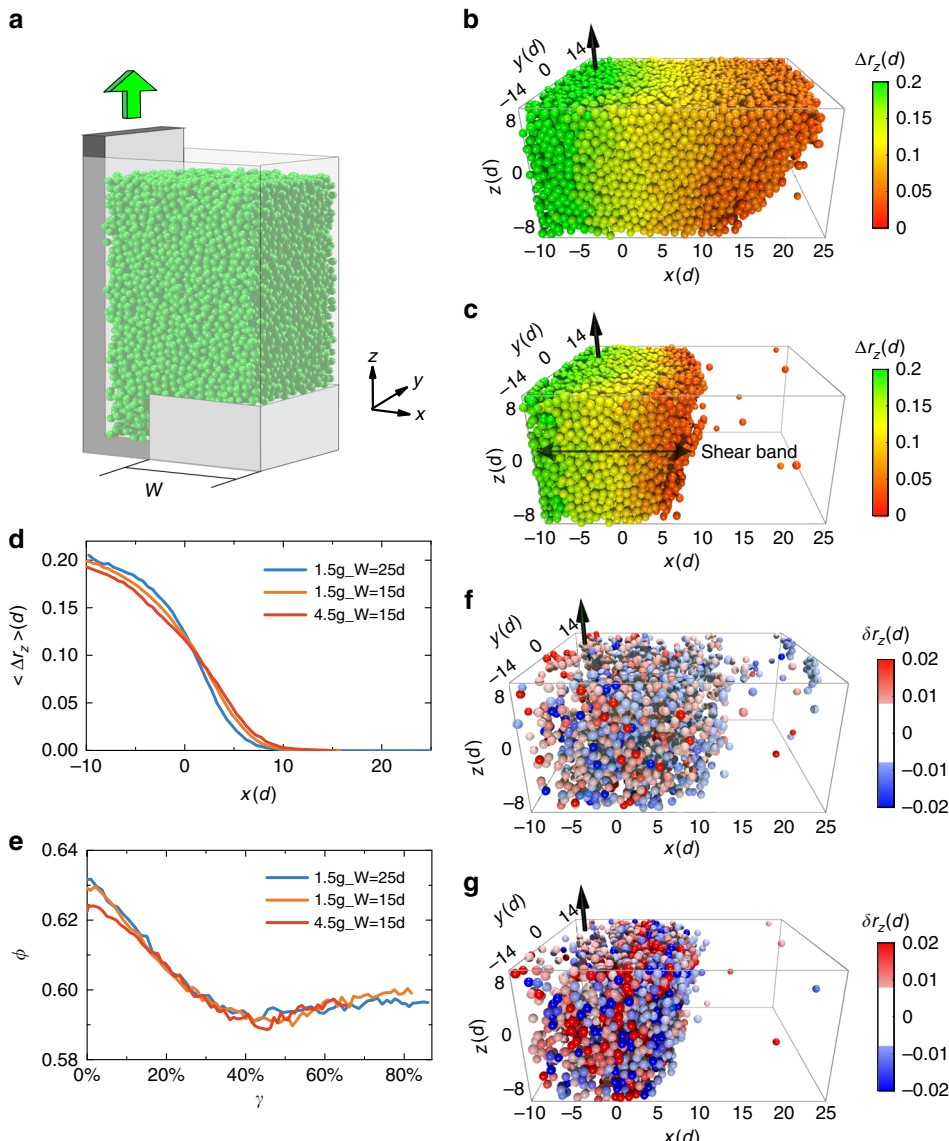

**Fig. 1** Macroscopic shear band and shear dilatancy. **a** Schematic of the plane shear cell. A roughly 2 mm-thick, 7 mm-wide granular particle slab was lifted up by the L-shaped bracket, exerting a shear stress on the bulk rest with a thickness of $W$. The origin of coordinate system is placed at the boundary between the slab and the bulk. **b, c** The absolute $z$-displacement $\Delta r_z$ of each particle (in units of particle diameter $d$) after a single shear step at the initial and the steady states, respectively. Particles with values smaller than $0.008d$ are not shown. **d** The average $z$-displacement profile along $x$-direction when the system is at the steady states. Three symbols correspond to three samples with different initial packing fractions $\phi$ and $W$. **e** The average volume fraction $\phi$ within the shear band decreases as the strain $\gamma$ increases, and it reaches steady state with $\phi = 0.590(5)$ after a critical strain around $\gamma = 40\%$. **f, g** The non-affine $z$-displacement $\delta r_z$ of each individual particle after a single strain step at the initial and the steady states, respectively. Particles with absolute values $|\delta r_z|$ smaller than $0.008d$ are not shown for clarity

are considered to belong to one cluster). From the distribution we can conclude that flip events are spatially localized since the cluster size is predominantly two (2–3 flip), three (3–2 flip) or four (2–2 flip), comprising only a single flip event. To analyze the occurrence probability of flip events, we calculated the flip frequency among all possible couples or triples in which we term any two coplanar tetrahedra as a couple and three coplanar tetrahedra as a triple. In a single shear step, only about 6% of couples or triples will flip and flips are more frequent in the shear band regime. Figure 2c shows the locations of the flipped couples and triples in the $x$–$z$ plane within a $2d$ thickness ($-1d < y < 1d$) after a shear step, overlaid with the corresponding non-affine displacement field at $y = 0$ when shear band is formed. The non-affine displacement field has been smoothed over a distance of two particle diameters. It is clear that correlation exists between

the flip sites and cores of large non-affine displacement regions[29]. Furthermore, we found that the orientational angles of flip events (Methods) are strongly correlated with the principal stress direction. Figure 2d shows the angular density distributions of the orientations of flipped tetrahedral groups with respect to the horizontal plane ($x$–$y$ plane). It is clear that unstable couples and triples have preferred orientations before and after the flips. Specifically, the couples with an orientation around $+45°$ are more likely to flip to form $-45°$ couples or $+45°$ triples, while the $-45°$ triples are more likely to flip to form $-45°$ couples. This is due to the fact that the particle distance tends to be compressed along the principal stress direction and expanded in the orthogonal direction which makes tetrahedral groups in specific orientations more vulnerable to flip instability. Once they flip, they have orientations which are difficult to flip again. We rule

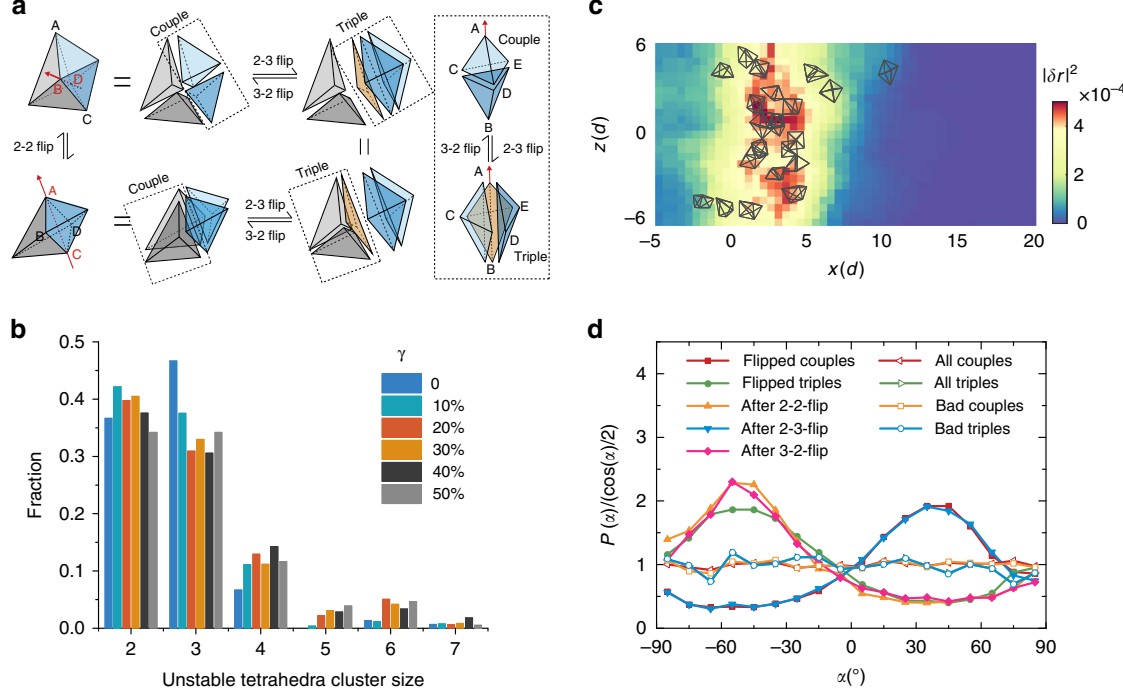

**Fig. 2** Properties of flip events. **a** Flip processes for unstable tetrahedra. 2–2 flip corresponds to a neighbor switching process in which a pair of couples exchange their vertices to form a new pair of couples (BD move closer to become neighbors while AC move away and cease to be neighbors). 2–3 (3–2) flip corresponds to a couple (triple) changing into a triple (couple). 2–2 flip can also be achieved by consecutive 2–3 and 3–2 flips. The orientation of a couple is defined by the red arrow as the axis direction of two non-coplanar vertices, and the orientation of a triple as the coaxial direction of the coplanar tetrahedra. **b** Histogram of size of unstable tetrahedral clusters after a single shear step at different $\gamma$. Face-adjacent unstable tetrahedra are considered to belong to the same cluster and the number of tetrahedra in the cluster is defined as the cluster size. **c** Flips in the $x$–$z$ plane within a $2d$ thickness ($-1d<y<1d$) region overlaid with non-affine displacement field at $y=0$. The non-affine displacement field has been smoothed over a distance of $2d$. **d** Probability density distributions of the orientation angles $\alpha$ of unstable couples or triples. $\alpha$ is the angle between the orientation of a couple or a triple and the horizontal plane ($x$–$y$ plane)

out the possibility that this anisotropy originates from the structural anisotropy of the system, since the angular distributions of all possible couples, triples or those formed only by bad tetrahedra ($\delta>0.245$, see below) are orientationally isotropic (Fig. 2d).

**Correlation between structure and plasticity.** We further investigated how topology change as characterized by flip events, structure change by tetrahedral shape parameter $\delta$, and plastic deformation by non-affine displacements $\delta\mathbf{r}$ are related among each other. We classify tetrahedra into three types based on their behaviors upon the application of a shear step (Fig. 3a): tetrahedra which flip (unstable tetrahedra), tetrahedra with none of their vertices involved in flip events (stable tetrahedra), and tetrahedra with some of their vertices involved in flip events owing to their spatial proximity to flip events (intermediate tetrahedra). First, we investigate how non-affine displacements of tetrahedra depend on their shape $\delta$ and flips. We characterize the non-affine displacements of a tetrahedron by defining its non-affine mobility $\mu=\frac{1}{4}\sum_{j=1}^{4}\delta r_j^2$, i.e., the mean square non-affine displacements of the four vertices of the tetrahedron. As shown in Fig. 3b, $\mu$ is largest for unstable tetrahedra, smallest for stable tetrahedra, and has values in between for intermediate tetrahedra, in agreement with the result of Fig. 2c. It is also interesting to note that $\mu$ has only a very weak dependency on the tetrahedral shape $\delta$ for stable tetrahedra. Since $\mu$ is directly related to the flip events and spatial proximity to them, we plot in Fig. 3c $\mu$ as a function of distance from unstable and stable tetrahedra. We recognize that unstable

tetrahedra correspond to the cores of large plastic activities and the stable tetrahedra correspond to cores of much weaker but still finite plastic activities. The two curves roughly merge around $r = 4d$, which yields the range of influence zone. Next we investigate the connection between the shapes of the tetrahedra and flip events. From Fig. 3d, it is obvious that flip is much more likely among highly distorted tetrahedra which have $\delta>0.245$. And the more distorted, the more likely a tetrahedron will flip in the subsequent shear step. Stable tetrahedra are more likely to have smaller $\delta$, i.e., more regular shape. Intermediate tetrahedra tend to have shapes between these two extremes. These results are reminiscent of previous findings in which a polytetrahedral glass order based on quasi-regular tetrahedra has been defined in granular packings based on $\delta>0.245$[20]. It is interesting to see that the defective structure associated with this order plays a significant role in plasticity, similar to the role played by dislocations in crystals. As shown in the following, the analogy is much more profound as we find that the flip process of unstable tetrahedra are equivalent to the rotation motions of 4-ring disclinations which are topological defects associated with rotational degrees of freedom (Supplementary Table 2). Although the shape parameter $\delta$ does not influence the non-affine displacements directly (Fig. 3b), it nevertheless yields a facilitation mechanism for subsequent plastic activities, i.e., a tetrahedron has to be heavily distorted to undergo a flip event upon shear, and it is therefore more likely have a new flip event close to a previous one since the tetrahedra are on average more distorted there (Supplementary Note 5). This facilitation effect is discernable within a diameter of $4d$ of the flipped site similar to the influence zone as observed in Fig. 3c, which is obtained by analyzing the spatial correlation of

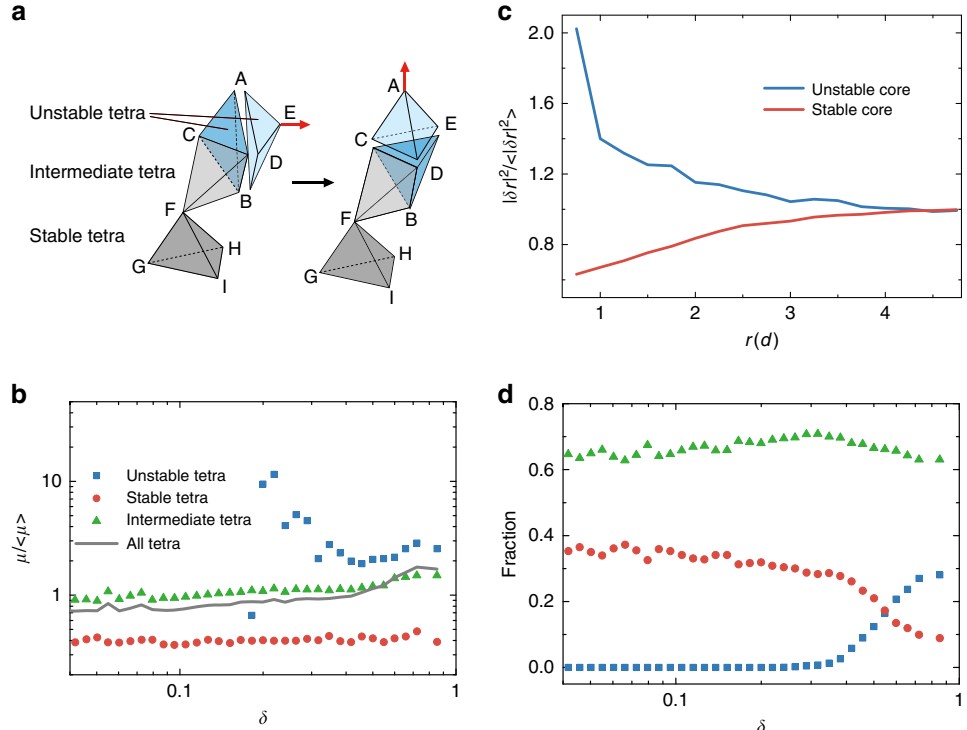

**Fig. 3** Tetrahedral species and their plastic properties. **a** Schematic of spatial relationship of the three types of tetrahedra. **b** Correlation between non-affine mobility $\mu$ and shape parameter $\delta$ for three types of tetrahedra. **c** Normalized square non-affine displacements of particles vs. distance $r$ from the unstable and stable cores. The unstable cores are geometrical centers of unstable tetrahedra. The stable cores are geometrical centers of stable tetrahedra. **d** Fraction of three types of tetrahedra as a function of the shape parameter $\delta$

flip events of two subsequent shear steps. The existence of facilitation mechanism thus indicates a spatial and temporal correlations between flip events. Recently, the collective behavior of microscopic plastic deformations in relation to macroscopic force fluctuations has been analyzed in sheared granular materials, and it is observed that a significant long-range strain correlation is directly related to the macroscopic avalanche behavior[29]. It is therefore interesting to investigate in the future how the local facilitation mechanism as identified here is related to shear band formation and avalanches[30, 31].

**Topological nature of plasticity.** It is well-known that for crystalline materials the carriers for microscopic plasticity are dislocations which are topological defects associated with translational degrees of freedom. For our system, we also characterized the topological nature of the flip events based on N-ring disclination structures[32]. An N-ring structure on Delaunay network represents a tetrahedral group with one edge as a common axis and coplanar between neighboring members. A 5-ring structure is considered to be the disclination-free ground state structure, whereas other N-ring structures possess disclination defects, which are topological defects associated with rotational degrees of freedom. The N-ring concept was originally developed to describe the potential ideal glass state. Since 5-ring structures alone cannot tile space, the ideal glass structure is conjectured to possess evenly spaced 6-ring structures in a 5-ring structure background as introduced by Frank and Kasper[32–34]. For low-density hard sphere systems, there should therefore exist many disclination defects. As shown in Fig. 4a, consistent with this picture, the 5-ring is most populous in the initial dense state, decreasing in amount after the shear, and the steady state has significantly more disclination defects. Interestingly, the only way an N-ring

structure can transform into another one is through flips. It turns out that 2–2 flip is equivalent to a 4-ring structure rotating into another 4-ring structure (Fig. 2a) and 2–3/3–2 flips correspond to a 4-ring structure transforming into a 3-ring structure and vice versa. Since a 3-ring structure (even more distorted tetrahedra) is mechanically very unstable and therefore only exists transiently, plasticity in our system essentially happens through 2–2 flips or the rotation of 4-ring structures. We emphasize that a 2–2 flip is therefore the only pathway for N-ring structure to transform between each other, e.g., Fig. 4b shows how a 2–2 flip process can change a neighboring 5-ring into a 4-ring structure. We therefore establish close connections between a 2–2 flip, rotation of a 4-ring structure, and local plasticity.

**Discussion**
In conclusion, we find that elementary plastic events in sheared granular materials mainly happen through flip events of highly distorted coplanar tetrahedra of the Delaunay network. This result supports the concept that highly distorted coplanar tetrahedra are structural defects of disordered granular packings and carriers of microscopic plasticity. Since flip events can also be described as the rotation motions of 4-ring disclinations which are topological defects associated with rotational degrees of freedom, close analogies with dislocations in crystals can be drawn. We believe our results should not be considered as applicable to granular materials only, but also to atomic and molecular amorphous systems, despite the fact that granular materials are athermal and have friction. To understand why the presence of friction does not modify the overall picture, it is useful to compare the potential/free energy landscape of a thermal glassy system with the one of a frictional granular system. These two landscapes can be expected to be quite similar on the

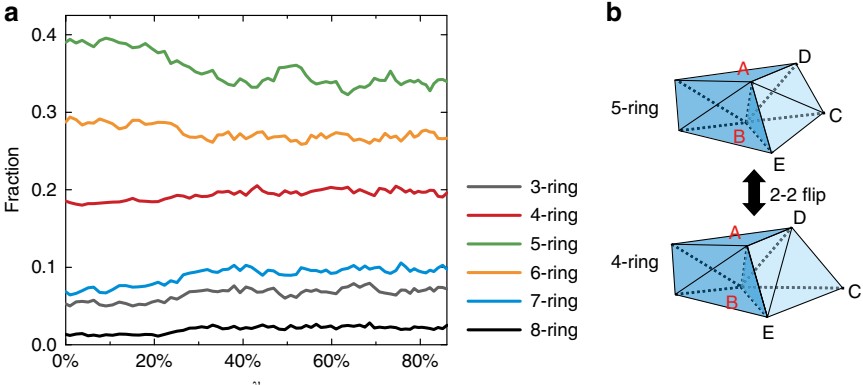

**Fig. 4** Evolution and transformation of N-ring structures under shear. **a** Evolution of the fraction of N-ring structures upon shear. **b** A 5-ring structure with AB as common axis, evolves into a 4-ring structure as the couple ABCE/ABCD forms a new couple DEBA/DEBC through 2–2 flip as AC and DE switch neighbors

length scale of the size of the particles. However, because of friction, the former will remain very rugged even on much smaller scales[35]; whereas, the landscape of an atomic-glass formers is basically smooth for length scales below the size of atoms. This difference in the landscapes will lead to rather different behavior in their plastic behaviors. Despite the fact that the topological pathways to a local saddle point in a granular material and a thermal glass will be quite similar, the microscopic dynamical and plastic behaviors of their two systems are quite different. For thermal glassy systems, the overcoming of the landscape barrier is related to thermal fluctuations and it will be instantaneous. In granular materials, on the other hand, since the pathway can be stabilized by friction, it can freeze the motion on the topological pathway of a plastic deformation followed in thermal glassy systems. So in general we expect that the structural and topological characteristics of plastic deformations as observed in our system will remain also valid in thermal glassy systems, i.e., our results should be applicable to a wide range of amorphous materials, thus allowing to gain insight into mechanical properties of such materials.

## Methods

**Experimental details**. A shear setup suitable for X-ray tomography study was built[27]. As shown in Fig. 1a, the setup is a rectangular acrylic glass container with dimension of $8(L) \times 7(W) \times 32(H)$ mm³. The shear is generated by a 2 mm-thick L-shaped bracket which can move in the vertical direction against a block of width $W$. The coordination axes are set so that the shearing direction is along $+z$-direction, opposite to gravity. The direction normal to the shear plane is the $x$-direction with coordination zero at the interface between the bracket and the block. Two blocks with $W = 25d$ and $W = 15d$ were used to investigate the influence of the boundary on shear band formation. The bracket surface and the opposite surface of the container were roughened by glued glass particles. Before each shear sequence, the particles are slowly poured into the container up to a height around 25 mm, then the container was tapped 10,000 times at two tapping intensity 1.5 g and 4.5 g to reach different initial steady state volume fraction $\phi$. Each tap consists of one cycle of 30 Hz sine wave at an interval of 0.5 s.

As shown in Fig. 1c, the initial packing was first recorded by a tomography scan. Then the shearing bracket was moved at a constant speed for around $1/3d$ before another tomography scan was carried out. Since each shear step has an effective strain of $\Delta\gamma = 1.2 \pm 0.2\%$ and lasts 0.525 s, the strain rate $\dot{\gamma}$ is 0.023 s⁻¹. The corresponding inertial number $I = \frac{\dot{\gamma}d}{\sqrt{P/\rho}}$ is about $3 \times 10^{-5}$, therefore ensuring the shearing quasi-static[36]. We estimate the pressure by $P = \phi\rho gH$, with the volume fraction $\phi = 0.60$, density $\rho = 2.7$ g cm⁻³, and the distance between the center of the imaging window to the upper surface of the packing is $H = 5$ mm.

The tomography imaging window has $2048(W) \times 560(H)$ pixels, with a spatial resolution of 6.5 μm/pixel, which can image about 3.6 mm-height section at the middle of the sample. A complete tomography scan consists of 1500 projection images, which takes about 5 min. Through image processing and particle tracking algorithms[20, 35], the centroids and trajectories of the all particles can be determined to be within an error less than $3.3 \times 10^{-3}d$.

**Data availability**. The datasets generated during and/or analysed during the current study are available from the corresponding author on reasonable request.

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

## Acknowledgements

The work is supported by the National Natural Science Foundation of China (No. 11175121 and 11675110).

## Author contributions

Y.W. designed the research. Y.C., J.L., B.K., C.X., Z.L., R.C., H.X., T.X., K.W., L.H., J.Z., and Y.W. performed the experiment. Y.C., J.L., W.K., and Y.W. analyzed the data and wrote the paper.

## Additional information

**Competing interests:** The authors declare no competing interests.

