## [Peer Review File · Nature Communications]

Reviewer #1 (Remarks to the Author):

The paper by Yixin Cao et al. presents an interesting X-ray tomography study of local structural (topological) changes in the plastic deformation of a granular material. Synchrotron X-ray tomography is used to track the structure of a granular material sheared within a band of thickness ~ 20 particle diameters, and to identify elementary flip events underlying the granular deformation. The authors find neighbour switching processes among coplanar tetrahedra of favourable orientation, and identify those as rotation motion of 4-ring disclinations, in analogy to dislocations in crystals.

The paper is sound and the data seems of high quality, in particular in view of the elaborate synchrotron tomography measurement technique. My central comment and only major criticism concerns the relation of the observed local topology changes to the growing Body of work on strain fields and avalanches in sheared granular and amorphous matter: there has been a lot of progress recently on understanding the emergent collective deformation and yielding of amorphous (granular) materials based on highly collective slip events (avalanches), and the current study should better connect to these recent developments. My feeling is that the identified tetrahedral flips sit at the core of previously identified strain- or non-affine field maxima, thus presenting an important origin of irreversibility; while the authors show this to some extent (in Fig. 3c), this connection should be made much clearer for the paper to have broad impact. Detailed comments are given below. If the authors can adequately respond to these issues, I'll be happy to recommend publication.

1. How do the locations of the 2-2 flips relate to the non-affine displacement field? The authors show this to some extent by plotting squared non-affine displacements in the neighbourhood of the flips in Fig. 3c, but that still doesn't allow for an overall picture. Can't they directly overlay the flip locations onto the non-affine displacement field, e.g. the flip locations in the slice in Fig. 2c onto non-affine displacement field (or strain field) in this slice? This would establish an immediate connection between the local changes studied in this paper, and the more collective/global view of other work. If the displacement field looks too scattered, some spatial averaging (smoothing) might help.
2. Related to the above: the displacement field shown in Figs. 1f,g comes very close to previously published work (internal imaging of a sheared granular material) by Denisov et al. (Nat. Comm. 7, 10641 (2016)), where also the connection to avalanches is established. The authors should cite this work, and put their work into perspective with this previously published work.
3. On page 6 bottom, the authors discuss a facilitation mechanism by which it is more likely to have a new flip event close to a previous one. Could this lead to avalanche formation? It just seems natural to discuss this relation as these avalanche phenomena have become such an important part of the literature of amorphous plasticity. The authors should discuss or speculate on this relation.
4. Minor comment: It took me some time to actually understand the tetrahedral flips in Fig. 2a. The reason is that the tetrahedra shown always have the same shape. I recommend the authors distort the tetrahedra according to the actual flips (i.e. change the A-C to B-D aspect ratio) to immediately make the topology changes clear.

Reviewer #2 (Remarks to the Author):

The manuscript reports the detailed structural evolution of granular materials under quasi-static shear using synchrotron X-ray tomography. The experimental results reveal that plastic deformations originate from flipping of structural defects, or unstable tetrahedra, under shear.

While the authors' experimental efforts and results are quite remarkable, I have concerns and questions about the manuscript, and I cannot recommend its publication in Nature Communications in its current state. My questions and concerns are outlined below:}

[1.] Novelty & applicability: The current manuscript follows a number of excellent papers on the topic of plastic deformations in a disordered solid. As mentioned in the manuscript, Shall and colleagues (Science 2007) experimentally showed the formation of localized shear transformation zones in colloidal glasses, while Amon et al (PRL 2012) visualized the appearance of microscopic defects in a granular packing under shear and elucidated their connection to the macroscopic rheology of the system. There also exists work by Nasuno et al (PRL 1997) who experimentally showed the microscopic rearrangement of grains that precede the slip of granular media under shear (not cited in the manuscript). Hence, given the amount of high-caliber work on the topic, the authors need to better substantiate the novelty and importance of their current findings.

From reading the manuscript carefully, the novel finding of the current manuscript is the characterization of local defects as 'unstable tetrahedra' that undergo flip events. While the experimental efforts and the results are remarkable, it is unclear what the physical significance of the new findings is, especially in light of the existing body of literature (I will comment on this more in the next bullet point). In addition, it is unclear from the manuscript how applicable the findings are to other materials. As the authors themselves point out towards the end of the manuscript, the large grain system is not governed by thermal fluctuations and has frictional contact between grains; therefore, the applicability of their results to other amorphous materials is questionable. Can the authors at least comment on the robustness of their results? Are their experimental findings largely independent of shear rate, as long as they are in the quasi-static regime? What about grain sizes, polydispersity and other boundary conditions? Overall, in my opinion, the novelty, applicability, and robustness of the current result are not made clear in the manuscript.

[2.] Physical interpretation: As mentioned above, the experimental efforts by the authors to experimentally measure the flip events are commendable. I understand that the experimental measurement itself is an impressive and important scientific discovery. However, in my opinion, for the results to be published in a high caliber journal such as Nature Communications, the manuscript needs more physical interpretation of the results.

What does the discovery of the defect structures actually say about the nature of the shear band formation? What does it add to the current understanding of the rheology of granular materials? Does it help prove or disapprove any existing hypothesis of the nature of microscopic plastic events in these systems? I believe that the convincing and well thought-out interpretation of the data is just as important as the data itself. For instance, the authors mention that the current discovery may be the 3D analogy to T1 events in 2D emulsions (or foams). What is the physical significance of this analogy? I am not asking that the authors must have quantitative theoretical models to explain or/and expand their data. However, the authors do not seem to offer any new physical insight about granular (or general disordered) media as a result of their experimental discovery, and I find this lack troubling. It is possible that I missed it in my reading, and I would be happy to be corrected.

[3.] Presentation: Overall, the manuscript is written in a clear, straight-forward way, but some parts could be improved. For instance, I identified some minor grammatical errors that need to be corrected (see Line 184 of the manuscript, for example). More importantly, I found some of the key figures to be not as clear. For instance, the authors state in the text that Fig. 1c is a clear example of the shear band formation, but perhaps to my untrained eye, I do not clearly see what the authors refer to as the shear band in the image. Simple labeling might have fixed this

problem. Similarly, the authors claim that Fig. 1g shows the local plastic events inside the shear band, which clearly differs from Fig. 1f. However, I cannot detect any significant difference between Fig. 1f and 1g. I would appreciate further clarification and, in general, more thoughtfulness in data representation in figures.

In summary, I believe this manuscript contains interesting results but is not appropriate for the publication in Nature Communications in its current state.

Dear Editor,

Please find enclosed the revised version of the manuscript “Structural and Topological Nature of Plasticity in Sheared Granular Materials” by Yixin Cao et al. which we have submitted for Nature Communications. The manuscript has been reviewed by two reviewers and both of them had a positive opinion on it while they also raise some questions. In the following we address the various questions/comments of the reviewers, all of them are constructive and helpful. These comments lead us to make several modifications to the paper which we think have improved its scientific value and hence we thank the reviewers for their work. We hope that the new version of the manuscript can now be considered as suitable for publications.

Reviewer #1 (Remarks to the Author):

The paper by Yixin Cao et al. presents an interesting X-ray tomography study of local structural (topological) changes in the plastic deformation of a granular material. Synchrotron X-ray tomography is used to track the structure of a granular material sheared within a band of thickness ~ 20 particle diameters, and to identify elementary flip events underlying the granular deformation. The authors find neighbour switching processes among coplanar tetrahedra of favourable orientation, and identify those as rotation motion of 4-ring disclinations, in analogy to dislocations in crystals.

The paper is sound and the data seems of high quality, in particular in view of the elaborate synchrotron tomography measurement technique.

Response: We thank the reviewer for this positive comment.

My central comment and only major criticism concerns the relation of the observed local topology changes to the growing Body of work on strain fields and avalanches in sheared granular and amorphous matter: there has been a lot of progress recently on understanding the emergent collective deformation and yielding of amorphous (granular) materials based on highly collective slip events (avalanches), and the current study should better connect to these recent developments. My feeling is that the identified tetrahedral flips sit at the core of previously identified strain- or non-affine field maxima, thus presenting an important origin of irreversibility; while the authors show this to some extent (in Fig. 3c), this connection should be made much clearer for the paper to have broad impact. Detailed comments are given below. If the authors can adequately respond to these issues, I'll be happy to recommend publication.

1. How do the locations of the 2-2 flips relate to the non-affine displacement field? The authors show this to some extent by plotting squared non-affine displacements in the neighbourhood of the flips in Fig. 3c, but that still doesn't allow for an overall picture. Can't they directly overlay the flip locations onto the non-affine displacement field, e.g. the flip locations in the slice in Fig. 2c onto non-affine displacement field (or strain field) in this slice? This would establish an immediate connection between the local changes studied in this paper, and the more collective/global view of other work. If the displacement field looks too scattered, some spatial averaging (smoothing) might help.

Figure R1. Flips in the x - z plane within a $2d$ thickness ($-1d < y < 1d$) region overlaid with nonaffine displacement field at $y=0$.

Response: We thank the reviewer for this suggestion. Fig. 2c of original manuscript only plotted the locations of the flip sites during one shear step. We now modified Fig. 2c as following the reviewer's suggestion by overlay the flip sites together with the nonaffine displacement field during one shear step when the shear band is formed. The figure shows x - z plane nonaffine displacement field at $y = 0$ and flips within thickness ($-1d < y < 1d$) region. The nonaffine displacement field has been smoothed over a distance of two particle diameters. It is clear that there indeed exists a correspondence between the flip sites and the cores of large plastic deformations. However, the correspondence is not as perfect as in Fig. 3 which average over all flip sites for all shear steps since during one shear step only about 6% of the couples flip even in the shear band regime. But despite the fluctuations that are present in Fig. 2C, the correlation is clearly visible. We have modified the text as well as the caption of Fig. 2 which now reads: *"Fig. 2c shows the locations of the flipped couples and triples in the x - z plane within a $2d$ thickness ($-1d < y < 1d$) after a shear step overlaid with the corresponding nonaffine displacement field at $y=0$ when shear band is formed. The nonaffine displacement field has been smoothed over a distance of two particle diameters. In a single shear step, only about 6% of couples or triples will flip and flips are more frequent in the shear band regime. It is clear that correlation exists between the flip sites and cores of large nonaffine displacement regions."*

2. Related to the above: the displacement field shown in Figs. 1f,g comes very close to previously published work (internal imaging of a sheared granular material) by Denisov et al. (Nat. Comm. 7, 10641 (2016)), where also the connection to avalanches is established. The authors should cite this work, and put their work into perspective with this previously published work.

Response: We thank the reviewer for this suggestion. Although in the current manuscript, our focus is on structural and topological nature of defect of disordered system and its relationship with elementary plastic deformation, it is indeed crucial to understand how their collective behaviors are related to macroscopic mechanical properties, especially the potentially important influence induced by the elastic interactions of the flipping sites. In Denisov et al. (Nat. Comm. 7, 10641 (2016)), a significant long-range strain correlation has been observed, which is correlated to the power-law distributed

macroscopic force fluctuations to establish the potentially universal critical yielding behavior of disordered solids. Although we didn't measure stress-strain curves and have no knowledge on the macroscopic avalanche size in the system, nevertheless, we have calculated the strain correlation similar to Denisov et al. as shown in Fig. R1. The significant anisotropic strain correlation is in very good agreement with the results (inset of Fig. 3a) of Denisov's. It is therefore very useful to combine the microscopic knowledge gained in the current study with Denisov's to establish a complete picture including microscopic plastic events to macroscopic avalanche behaviors.

To put the current work in a bigger perspective of recent research activities on the yielding and plastic behaviors of disordered materials including granular materials, we notice that one major research focus is on the collective avalanche behaviors induced by the elastic interactions among the elementary plastic events. This has been pursued by means of a mean-field depinning-like (Dahmen et al. Nature Phys. 7, 554 (2011)) or as recently by an Eshelby-like interaction approach (Muller and Wyart, Annu. Rev. Condens. Matter Phys 6, 177 (2015)). Therefore, a thorough investigation of the interactions between the elementary plastic events is crucial. However, experimentally to differentiate a mean-field mechanical noise temperature (depinning-like) from a signed noise temperature (Eshelby-like) as induced by the flipping sites remains difficult. It will also be interesting to test whether the long strain field correlation is directly responsible for the avalanches.

Figure R2. Correlation function of the fluctuations of shear strain \mathcal{E}_{xz}

3. On page 6 bottom, the authors discuss a facilitation mechanism by which it is more likely to have a new flip event close to a previous one. Could this lead to avalanche formation? It just seems natural to discuss this relation as these avalanche phenomena have become such an important part of the literature of amorphous plasticity. The authors should discuss or speculate on this relation.

Response: We thank the reviewer for this suggestion. As shown in Fig. R2, we calculated the probability of finding a flip at $t_0 + \Delta t$ at distance r from flip centers which happen at t_0 and compared it with the value obtained if one assumes that the flip events are spatially totally uncorrelated. As it is obvious from the figure, there clearly exists a slight facilitation effect within the influence zone

size (about $4d$) of the plastic core which is consistent with the information as conveyed by Fig. 3c. Interestingly, the facilitation mechanism is rather short-ranged as it only tends to facilitate potential sites in its close neighborhood. The short-range nature of the facilitation could be due to the rather low pressure of the system since we have an open container. Whether this short-range facilitation can induce a global avalanche behavior remains an interesting research topic for the future. We have modified the text as follows: *“This facilitation effect is noticeable within the same influence zone range as identified in Fig. 3c, which is obtained by analyzing the spatial correlation of flip events of two subsequent shear steps. The existence of facilitation mechanism suggests spatial and temporal correlations between flip events. Recently, the collective behavior of microscopic plastic deformations in relation to macroscopic force fluctuations has been analyzed in sheared granular materials, and it is observed that a significant long-range strain correlation is responsible for the macroscopic avalanche behavior. It is therefore interesting to investigate in the future whether the facilitation mechanism is related to the strain correlation in certain way and can lead to macroscopic avalanche behavior.”* We have also added the references Denisov et al. (Nat. Comm. 7, 10641 (2016)), Müller, M. et al. (Annu. Rev. Condens. Matter Phys. 6, 177-200 (2015)) and Dahmen, K. A. et al. (Nat. Phys. 7, 554 (2011)).

Figure R3. Normalized probability of flip events at a radial distance r from a previous one.

4. Minor comment: It took me some time to actually understand the tetrahedral flips in Fig. 2a. The reason is that the tetrahedra shown always have the same shape. I recommend the authors distort the tetrahedra according to the actual flips (i.e. change the A-C to B-D aspect ratio) to immediately make the topology changes clear.

Response: We thank the reviewer for this comment. We modified the figure to make it more reader-friendly. In the lower panel of Fig. 2a, we changed the A-C to B-D aspect ratio to make the topology changes clearer.

Reviewer #2 (Remarks to the Author):

The manuscript reports the detailed structural evolution of granular materials under quasi-static shear using synchrotron X-ray tomography. The experimental results reveal that plastic deformations originate from flipping of structural defects, or unstable tetrahedra, under shear. While the authors' experimental efforts and results are quite remarkable, I have concerns and questions about the manuscript, and I cannot recommend its publication in Nature Communications in its current state. My questions and concerns are outlined below: }

Response: We thank the reviewer for the positive comments on our experimental results. As for the reviewer's concerns and questions, we will address them in the following.

[1.] Novelty & applicability: The current manuscript follows a number of excellent papers on the topic of plastic deformations in a disordered solid. As mentioned in the manuscript, Shall and colleagues (Science 2007) experimentally showed the formation of localized shear transformation zones in colloidal glasses, while Amon et al (PRL 2012) visualized the appearance of microscopic defects in a granular packing under shear and elucidated their connection to the macroscopic rheology of the system. There also exists work by Nasuno et al (PRL 1997) who experimentally showed the microscopic rearrangement of grains that precede the slip of granular media under shear (not cited in the manuscript). Hence, given the amount of high-caliber work on the topic, the authors need to better substantiate the novelty and importance of their current findings.

From reading the manuscript carefully, the novel finding of the current manuscript is the characterization of local defects as 'unstable tetrahedra' that undergo flip events. While the experimental efforts and the results are remarkable, it is unclear what the physical significance of the new findings is, especially in light of the existing body of literature (I will comment on this more in the next bullet point). In addition, it is unclear from the manuscript how applicable the findings are to other materials. As the authors themselves point out towards the end of the manuscript, the large grain system is not governed by thermal fluctuations and has frictional contact between grains; therefore, the applicability of their results to other amorphous materials is questionable.

Response: We thank the reviewer for these comments. From our understanding, the reviewer asked two questions. First is on the novelty of our work as it stands amongst previous works, at least with the three works mentioned. We are sorry that the way we wrote the paper didn't make it clear to the reviewer on what's the core message we want to deliver.

To make a long story short, in the current work, we believe we found the structural defect of disordered granular packings which bears great analogies with its counterpart of dislocations in crystalline solids. We have included a table in the following to make this analogy more straightforward. This table is also included in supplemental materials to help the reader understand the analogy more directly.

	Crystalline solid	Disordered granular packings
Structure order	crystalline order	glass order (regular tetrahedra)
Structure defect	Dislocation	highly distorted tetrahedral (4-ring disclination)
Plastic event	translational motion of dislocation	flip events (rotation of 4-ring disclination)

As for the novelty of current work as compared to previous ones. The work by Schall et al. (Science 318, 1895 (2007)) essentially established that shear transformation zones (STZ) are localized in space and they can potentially be characterized as defects. However, from the very beginning, a STZ is characterized as the “consequence” instead of a “cause” of a plastic event. Therefore, the structural and/or topological nature of the defects which are responsible for plastic deformations in disordered solids has never been clearly identified, in contrast to the situation in crystalline materials for which the dislocation takes up the role (which is a structural defect of a crystalline order, a topological defect of translational motion, and the bearer of local plasticity). In the current work, we want to follow the more traditional material science approach, i.e. the philosophy that “structure determines property” by establishing that the highly distorted tetrahedra (four-ring disclination) are structural defects of an amorphous structural order, a topological defect of rotational motion, and bearer of a local plasticity in disordered solids. This message is clearly absent in previous works, and it is therefore an important step forward in the understanding of the mechanical behaviors of granular materials, and more generally disordered solids.

As for Amon et al. ’s scattering work (Phys. Rev. Lett. 108, 135502 (2012)) which established the connection between macroscopic strain-stress curves with the local plastic events. It essentially addresses the important question on how one can account for macroscopic mechanical behaviors of disordered solids from the collective behavior of elementary plastic events, which mirrors Denisov et al. (Nat. Comm. 7, 10641 (2016))’s work as suggested by reviewer #1, which is not the main focus of the current paper.

Regarding the work by Nasuno et al.’s (Phys. Rev. Lett. 79, 949 (1997)) on the imaging of plastic events on sheared granular layers: These early studies on the motion of the grains under shear have given important qualitative insight, but at that time it was not possible to make a *quantitative* analysis. For such an analysis that permits a detailed characterization of a plastic event in 3D one needs a very advanced technique like X-ray tomography (used by us) or index matching method similar to Denisov et al. (Nat. Comm. 7, 10641 (2016)). So we feel that our present results are of a very different nature than the ones by Nasuno et al.’s.

As for the universality for the results of our current work. We are sorry if our paper insinuated the (wrong) idea that granular materials have friction which are absent in real glassy materials will make the story very different. To understand whether our results can be applied to general disordered solids and what role friction plays in the existing picture, we introduce a potential/free energy landscape picture in which one can clarify more easily the similarity and difference between a thermal glassy system and a frictional granular system. Due to the presence of friction, the landscape of granular systems can be seen to be rather similar to a glassy landscape on length scales of the size of the

particles, but it remains very rugged even on much smaller scales, whereas the landscape of atomic-glass formers are basically smooth for length scales below the size of the atoms. This difference in the landscapes will lead to rather different behavior in their plastic behaviors. Despite the fact that the topological pathways to a local saddle point will be quite similar in a granular material and a thermal glass, the microscopic dynamical behaviors of these two systems are quite different. For thermal glassy systems, the overcoming of the landscape barrier is related to thermal fluctuations and it will be instantaneous. In granular materials, on the other hand, since the pathway can be stabilized by friction, it can freeze the motion on the topological pathway of a plastic deformation followed in thermal glassy systems. So in general we expect that the structural and topological characteristics of plastic deformations as observed in our system will remain also valid in thermal systems. We have modified the text in the conclusion as follows: *“We believe our results should not be considered as applicable to granular materials only, but also are general for atomic and molecular amorphous systems despite the fact that granular materials have friction. To understand why the presence of friction does not modify the overall picture, we compare the potential/free energy landscape between an atomic-glass former and a frictional granular system. Due to the presence of friction, the potential/free energy landscape of a granular system can be seen to be rather similar to that of an atomic-glass former on length scales of the size of particles, but it remains very rugged even on much smaller scales, whereas the landscape of an atomic-glass formers is basically smooth for length scales below the size of atoms. This difference in the landscapes will lead to rather different behavior in their plastic behaviors. Despite the fact that the topological pathways to a local saddle point will be quite similar between these two systems, the microscopic dynamics will be quite different. For atomic-glass former, the overcoming of the landscape barrier is owing to thermal fluctuations and it will be almost instantaneous. In granular materials, on the other hand, since the pathway can be stabilized by friction, it can freeze the motion on the topological pathway of a plastic deformation followed in atomic-glass formers. Therefore we expect that the structural and topological characteristics of plastic deformations as observed in our system will remain also valid in thermal glassy systems, and our results can bring great insights to the understanding of the mechanical properties of general amorphous materials.”*

Can the authors at least comment on the robustness of their results? Are their experimental findings largely independent of shear rate, as long as they are in the quasi-static regime? What about grain sizes, polydispersity and other boundary conditions? Overall, in my opinion, the novelty, applicability, and robustness of the current result are not made clear in the manuscript.

Response: The reviewer is right that our results should be universal as long as it is in the quasi-static regime. As for large shear rate, we suspect that it will be the same, but since we don't have the experimental data on it, this is just our guess. So at current stage, we can only speculate on it. Regarding the grain size, polydispersity and boundary condition effects, we believe that grain size is not a major concern since we have already macroscopic sized grains. As for polydispersity, despite that our system is close to be monodisperse, the topological nature of the 4-ring characterization will be robust even for relatively polydisperse systems (similar scenario to the original bubble raft experiment). However, how to characterize the local order might have to be slightly modified since it has to correspond to a local dense packing and a local energy minimum in energy landscape. However, this

difficulty is not that much different from the one encountered in crystalline materials in which many different structural orders have to be defined. As for the boundary effect, since in our current work, we have mainly focused on the shear band regime and we have intentionally modified the container shape to access the boundary influence on our results, so we are confident that boundary does not play a significant role on our general conclusions.

[2.] Physical interpretation: As mentioned above, the experimental efforts by the authors to experimentally measure the flip events are commendable. I understand that the experimental measurement itself is an impressive and important scientific discovery. However, in my opinion, for the results to be published in a high caliber journal such as Nature Communications, the manuscript needs more physical interpretation of the results.

What does the discovery of the defect structures actually say about the nature of the shear band formation? What does it add to the current understanding of the rheology of granular materials?

Response: In our work, we establish the nature of the elementary plastic events in granular materials, insight that so far does not exist. Since our focus is on the microscopic behavior, it will be very useful to use this knowledge in order to gain a better understanding of the mesoscale behaviors and also to macroscopic rheological measurements. Hence our results should allow to establish a more rigorous microscopic basis for existing empirical rheological models and the formation of shear bands, which is also suggested by Reviewer 1. This is the same route we have adopted for the development of liquid state theory back in 70s.

Does it help prove or disapprove any existing hypothesis of the nature of microscopic plastic events in these systems? I believe that the convincing and well thought-out interpretation of the data is just as important as the data itself. For instance, the authors mention that the current discovery may be the 3D analogy to T1 events in 2D emulsions (or foams). What is the physical significance of this analogy? I am not asking that the authors must have quantitative theoretical models to explain or/and expand their data. However, the authors do not seem to offer any new physical insight about granular (or general disordered) media as a result of their experimental discovery, and I find this lack troubling. It is possible that I missed it in my reading, and I would be happy to be corrected.

Response: We believe that our results overall is consistent with previous results although it gives a more complete picture on the nature of plasticity and structural defects in disordered solid. To give the reviewer some idea what we mean, let's look at the T1 event in 2D. Originally the T1 plastic event in 2D is considered as a simple neighbor switching event. However, using the language developed in the present work, we can interpret the T1 event as the only topological pathway for a pair of Delaunay triangles to get destroyed, and only heavily distorted Delaunay triangles can be destroyed. The neighbor switching is also a rotation of a two-dimensional disclination, then similar defect structure argument can also be applied to 2D also. Therefore, our work offers a universal interpretation of plastic behavior in both 2D and 3D based on some structural defects, an insight that we consider as an important step forward in our understanding of the dynamics of driven granular materials.

[3.] Presentation: Overall, the manuscript is written in a clear, straight-forward way, but some parts could be improved. For instance, I identified some minor grammatical errors that need to be corrected (see Line 184 of the manuscript, for example). More importantly, I found some of the key figures to be not as clear. For instance, the authors state in the text that Fig. 1c is a clear example of the shear band formation, but perhaps to my untrained eye, I do not clearly see what the authors refer to as the shear band in the image. Simple labeling might have fixed this problem. Similarly, the authors claim that Fig. 1g shows the local plastic events inside the shear band, which clearly differs from Fig. 1f. However, I cannot detect any significant difference between Fig. 1f and 1g. I would appreciate further clarification and, in general, more thoughtfulness in data representation in figures.

Response: We thank the referee for the comments, we have modified the figures and made the corrections. We changed the colorbar of Fig. 1b-c to make the shear band in Fig. 1c more obvious and we also labeled it. In Fig. 1g, there are more spheres which has large $|\delta r_z|$ (where color blue/red is darker) than that of Fig. 1f, and also the spheres which has large $|\delta r_z|$ is more concentrated in Fig. 1g, which is a sign of shear localization and shear band formation.

In summary, I believe this manuscript contains interesting results but is not appropriate for the publication in Nature Communications in its current state.

We hope that with the modification suggested by the reviewer the message of the manuscript has become clearer and hence will be judged to be appropriate to be published in Nature Communications.

Dear Editor,

Please find enclosed the final version of the manuscript “Structural and Topological Nature of Plasticity in Sheared Granular Materials” by Yixin Cao et al. which we have submitted for Nature Communications. The manuscript has been reviewed by two reviewers and neither of them raised any issues to be addressed.

REVIEWERS' COMMENTS:

Reviewer #1 (Remarks to the Author):

The authors have convincingly addressed my comments. In particular, the connection to the work on strain correlations and avalanches is now well established. In my view, this puts the paper nicely in context with published work, and makes the novel findings of the current studies clear. I hence recommend publication.

Reviewer #2 (Remarks to the Author):

The authors have thoughtfully addressed my concerns in their report, and the revised manuscript has much improved in clarity and content. I do not have any additional comments or concerns about the revised manuscript.